# An Algorithm Combining Patient Performance Status, Second Hit Analysis, PROVEAN and Dann Prediction Tools Could Foretell Sensitization to PARP Inhibitors in Digestive, Skin, Ovarian and Breast Cancers

**DOI:** 10.3390/cancers13133113

**Published:** 2021-06-22

**Authors:** Sandy Chevrier, Corentin Richard, Thomas Collot, Hugo Mananet, Laurent Arnould, Romain Boidot

**Affiliations:** 1Department of Biology and Pathology of Tumors, Georges-François Leclerc Cancer Center-UNICANCER, 21079 Dijon, France; schevrier@cgfl.fr (S.C.); crichard@cgfl.fr (C.R.); larnould@cgfl.fr (L.A.); 2Department of Medical Oncology, Georges-François Leclerc Cancer Center-UNICANCER, 21079 Dijon, France; tcollot@cgfl.fr; 3Platform of Transfer in Cancer Biology, Georges-François Leclerc Cancer Center-UNICANCER, 21079 Dijon, France; hmananet@cgfl.fr; 4UMR CNRS 6302, University of Burgundy, 21079 Dijon, France

**Keywords:** VUS, homologous recombination, PARP inhibitors, response, progression-free survival

## Abstract

**Simple Summary:**

PARP inhibitors, a family of targeted cancer therapeutics, have been shown to be efficient in patients with some deficiencies in the homologous recombination machinery. However, a quick and reliable identification of patients who would benefit from such therapies remains a challenge. In particular, patients with tumors carrying variants of unknown significance (VUS) in homologous recombination genes do not currently benefit from PARP inhibitor treatments. In this study, we present an algorithm that may allow classification of these variants with regard to their impact on tumor responsiveness to PARP inhibitors. If validated on a larger patient sample, our algorithm would allow patients with tumors potentially responsive to PARP inhibitors to benefit from this therapy.

**Abstract:**

PARP inhibitors yield interesting outcomes for patients with ovarian tumors harboring *BRCA1* or *BRCA2* mutation, but also with other tumors with homologous repair (HR) deficiency. About 40% of variants are variants of unknown significance (VUS), blocking the use of PARP inhibitors. In this study, we analyzed NGS data from 78 metastatic patients treated with PARP inhibitors. We tested NGS data and in silico predictions to classify VUS as potentially benign or deleterious. Among 41 patients treated with olaparib, three had tumors harboring benign and 26 pathogenic variants, while 12 had VUS. Progression-Free Survival (PFS) analysis showed that benign variants did not respond to olaparib whereas pathogenic variants were associated with a median PFS of 190 days. Surprisingly, median PFS of patients with VUS-carrying tumors suggested that some of them may be sensitive to PARP inhibitors. By testing different in silico predictions and variant allelic frequency, we obtained an algorithm predicting VUS sensitivity to PARP inhibitors for patients with a Performance Status below 3. Our work suggests that VUS in HR genes could be predicted as benign or deleterious, which may increase the number of patients eligible for PARP inhibitor treatment. Further studies in a larger sample are warranted to validate our prediction algorithm.

## 1. Introduction

The development of targeted therapies has revolutionized the management of cancer patients. Recently, the base excision repair (BER) pathway has become a new therapeutic target. Poly(adenosine diphosphate)ribose polymerase (PARP) inhibitors have emerged; the first one to be approved was Olaparib, which yielded spectacular results in patients with stage III-IV ovarian cancer sensitive to platinum salts [1]. The mechanisms of action of PARP inhibitors are based either on enzymatic inhibition of PARP or on PARP trapping [1]. Their efficacy is largely linked to the presence of deficiencies in the homologous recombination pathway, and in particular the presence of *BRCA1* or *BRCA2* pathogenic mutations. These findings led to the approval of olaparib for the treatment of patients with progressive metastatic ovarian tumors sensitive to platinum salts and harboring somatic or germline *BRCA1* or *BRCA2* mutations. Recently, olaparib has also been approved as frontline maintenance therapy for the treatment of ovarian cancer patients in Europe, based on the results of the SOLO-I study [2]. Furthermore, two other PARP inhibitors: niraparib and rucaparib, have been granted marketing approvals for the treatment of patients with platinum-sensitive ovarian cancer [3,4]. PARP inhibitors have also been tested for the treatment of cancers at other sites, such as breast [5,6], pancreatic [7], and prostate cancer [8].

Concerning olaparib use in Europe, clinical use approval indicates that prescription should be limited to cases with a pathogenic or probably pathogenic variant (as defined by oncogenetic criteria), and sensitivity to platinum salts. The pathogenicity of germline *BRCA1* and *BRCA2* variants is established by immunoprecipitation functional assays, Western blotting, gamma irradiation, comparative structural modeling [9], as well as by segregation studies linking the presence of family susceptibility to breast and ovarian cancer in patients carrying these genetic variants [10,11]. These analyses are long, and only enable variants to be classified at a late stage. As such, they are not suited for integration in the clinical management of patients suffering from progressing cancer, in whom a variant of unknown significance (VUS) in a homologous recombination gene, such as *BRCA1* or *BRCA2*, is discovered. More recently, in silico tools for predicting variant pathogenicity have been developed. These tools are based on mathematical calculations evaluating the probability of amino acid impact on the protein structure. In parallel, as homologous recombination genes are tumor suppressors, both alleles of the gene need to be altered (the so-called ‘double hit’). The development of high-throughput sequencing (panel, exome, and genome sequencing) has led to a substantial increase in detection of VUS, which may represent around 40% of all variants [12]. This indicates that current clinical recommendations may miss up to 40% of patients who may benefit from treatment with PARP inhibitors, especially olaparib, according to clinical recommendations. In order to offer patients a rapid and reliable therapeutic choice, it is necessary to have a tool to quickly assess the impact of VUS detected in tumors.

In this retrospective study, we aimed to design an algorithm allowing one to predict the responsiveness of VUS-harboring tumors to PARP inhibitors as soon as molecular biology data are available, using the data obtained by Next Generation Sequencing from gene panel, exome, or genome sequencing.

## 2. Materials and Methods

### 2.1. Ethics Approval and Consent to Participate

This study on patient samples was conducted in accordance with the Declaration of Helsinki and approved by the Ethics Committee of the Georges-François Leclerc Cancer Center (Dijon, France) under the number 00010311, and by the Consultative Committee of Burgundy (Dijon, France) for the Protection of Persons Participating in Biomedical Research (Comité Consultatif de Protection des Personnes en Recherche Biomédicale de Bourgogne). Written informed consent was obtained from all patients. Patients for whom tumor and germline exome sequencing data were available were enrolled in the EXOMA clinical trial [13], and registered with ClinicalTrials.gov under the identifier NCT02840604.

### 2.2. DNA Extraction

DNA was extracted from formalin-fixed paraffin-embedded (FFPE) tumor specimens, five 8-µm tumor slices per patient, as previously described [14]. DNA from whole blood (germline DNA) was isolated using the Maxwell 16 Blood DNA Purification Kit (Promega) following the manufacturer’s instructions. Quantity of extracted genomic DNA was assessed using a fluorimetric method and a Qubit device.

### 2.3. Exome Sequencing

Libraries were constructed from 200 ng DNA sheared with a Covaris device to obtain fragments of about 300 bp by using SureSelect Human All Exon v6 kit (Agilent Technologies, Santa Clara, CA, USA), following the manufacturer’s protocol. Paired-end (2 × 111 bases) sequencing was performed on a NextSeq500 device (Illumina). Samples were multiplexed to obtain a mean coverage between 80 and 100×. Obtained sequences were aligned and annotated with the human Hg19 genome based on the SureSelect Human all Exon v6 manifest using BWA and GATK algorithms. Only sequences with a read depth of 10×, a mutation allele frequency greater than 5%, and a frequency below 1% in the general population were kept for further analysis.

### 2.4. Complex Analyses

All complex analyses of the exome data were performed following software developers’ instructions. Mutational signatures [15] data was generated using DeconstructSigs (v1.8.0) [16]. Tumor Mutational Burden (TMB) was calculated using the number of significant single nucleotide variants (SNV; untranslated regions, synonyms, introns, and intergenic SNVs were filtered out) divided by the number of megabases covered (with depth greater than 7× for blood samples and greater than 14× for tumor samples). Sample base coverage was obtained with the DepthofCoverage tool from the GATK (v3.6) [17,18,19]. Neoantigen information was generated using the pVACseq tool from the pVACtools suite (v1.1.5) as described in [20]. Small deletions were extracted from somatic variant files (vcf), deletions and losses of heterozygosity (LOH) greater than 10 Mb were obtained using Titan (v1.23.1) [21]. The homologous recombination deficiency (HRD) score was assessed by following a previously published pipeline [22].

### 2.5. Statistical Analysis

All statistical analyses were performed with GraphPad Prism version 8.3. Progression-Free Survival (PFS) significance was determined by the Kaplan–Meier method using the log rank (Mantel-Cox) test. Endpoint of PFS was progression of disease or death. Obtained *p*-values, whatever the significance, are indicated in the figures. Of note, multiple testing was performed, which might have led to false positive results, especially given the small size of the population studied.

## 3. Results

### 3.1. Patient Characteristics

From the 2 August 2015 to the 27 September 2017, 41 metastatic patients from the Georges-François Leclerc Center were treated with olaparib (Table 1). Among these, 23 (56.2%) had ovarian cancer (22: high grade serous carcinoma and 1: unknown histology), eight (19.5%) had breast adenocarcinoma, eight (19.5%) had adenocarcinoma of the digestive tract (five of the pancreas, two colon and one rectum), one (2.4%) had a clear cell carcinoma of the endometrium, and one (2.4%) had basal cell carcinoma of the skin. Among the 41 patients, 39 (92.9%) received platinum-based chemotherapy before olaparib treatment, with a median progression-free survival of 126 days (Figure 1A).

An analysis by site of origin showed that patients with digestive tract tumors were treated longer with platinum salts (median PFS of 212.5 days), followed by those with ovarian tumors (126 days), breast tumors (122 days), endometrial tumor (108 days), and skin carcinoma (63 days) (Figure 1B). The 41 patients treated with olaparib had a median PFS of 136 days (Figure 1C), with a high heterogeneity between cancer sites. Indeed, in our cohort, tumors of the digestive tract seemed largely resistant to olaparib treatment (PFS of 31.5 days), followed by breast tumors (167 days), endometrial tumor (190 days), skin carcinoma (210 days), and ovarian tumors (256 days) (Figure 1D). It is established that response to PARP inhibitors is correlated with clinical response to platinum salts. In our study group, out of eight patients with a digestive tract cancer, seven responded to platinum salt treatment: two patients achieved partial response and five achieved stable disease. Only one patient did not respond to platinum salt. Despite this fact, only one patient had a partial response to olaparib but died due to a non-related (non-cancer) cause (Figure 1E). All these patients were treated with olaparib as a new line of treatment after they progressed under previous treatment. On the eight breast cancer patients in our cohort, six received platinum-based treatment after several lines of treatment with anthracyclins, taxanes, 5-fluorouracil, and eribulin, and responded to the platinum treatment (two achieved a complete response, three achieved a partial response and one achieved stable disease). All six received olaparib as maintenance treatment. The two other patients received olaparib as a new line of treatment after progression under anthracycline-based treatment. Overall, out of the eight patients receiving olaparib treatment, three had rapid disease progression (one of whom had experienced a complete response to platinum, and another one a partial response), three achieved a partial response, one achieved stable disease, and one achieved a complete response that is still ongoing (Figure 1F). Concerning the 23 ovary cancer patients, all were treated by platinum salt and then maintained by olaparib. All patients except one showed a response to platinum (three: a complete response, fifteen: a partial response, and four: stable disease). The last one did not respond to platinum at all. When administered olaparib treatment, eight patients experienced a complete response, three had a partial response, three had stable disease, eight had progressive disease and one patient had to stop the treatment due to an allergic reaction (Figure 1G). Concerning patients with cancers at the two other sites, both responded to platinum: one with stable disease and one with a complete response. Both also showed a response to the olaparib they received as a new line of treatment: one with stable disease and one with a partial response (Figure 1H). Finally, in our whole population, it appeared that progression-free survival under olaparib treatment was significantly associated with response to platinum treatment (Figure 1I).

### 3.2. Some VUS May Respond to Olaparib

Among 41 patients treated with olaparib, three had tumors harboring benign variants, 12 had a VUS, and 26 had a pathogenic variant in homologous recombination genes (Table 2). Comparing survival of patients treated with olaparib stratified by the three variant classes present in their tumors, we found that patients with tumors carrying a benign variant had the shortest median PFS (81 days), whereas those with tumors harboring a pathogenic variant had the longest PFS (median 190 days), while patients with VUS-carrying tumors had intermediate PFS (median 127 days) (Figure 2A), implying that the variant class is associated with response to olaparib. Focusing on VUS and pathogenic variants only, we found that the median PFS for patients with *BRCA2* variants (259.5 days) was higher than for those with variants of *BRCA1* (152 days) and of other genes (190 days), but this difference was not statistically significant (Figure 2B). Moreover, some variants were associated with PFS over 120 days (which was the maximum PFS observed for patients with tumors harboring benign variants), regardless of whether the genes harbored a VUS or a pathogenic variant (Figure 2B). This suggests that these variants could sensitize tumors to olaparib. Furthermore, we observed that some patients with *BRCA1* or *BRCA2* pathogenic variants had longer PFS than patients with VUS-carrying tumors (Figure 2C). Nevertheless, some patients with tumors carrying pathogenic variants did not respond to olaparib (PFS < 120 days), whereas those with some VUS had PFS over 120 days (Figure 2C). Concerning variants in other genes, tumors with VUS seemed to have the same behavior in response to olaparib as those with pathogenic variants (Figure 2D). Having analyzed the distribution of pathogenic variants and VUS by cancer site we found that pathogenic variants of *BRCA1* and *BRCA2* were present only in ovarian and breast tumors, and those of other genes were present in four different organs (ovary: 1, breast: 1, endometrium: 1 and digestive tract: 2), whereas VUS were present in all cancer sites (Figure 2E).

### 3.3. Response of VUS to Olaparib Could Be Predicted before Treatment

Since response to PARP inhibitor was described as correlated with response to platinum salt, we wondered whether this was the case for the 12 patients with VUS-carrying tumors in our study group. Based on best response observed under platinum treatment (Appendix A), we found that sensitivity to platinum salt was not statistically significantly correlated with a longer PFS in olaparib-treated patients (Appendix A), even though a trend could be observed (*p* = 0.1649).

With the development of Next Generation Sequencing enabling sequencing of homologous recombination panels, exomes or genomes, data obtained include quality and quantity information, such as the type of mutation and allelic frequency of alternative variants. Bioinformatics analyses also make it possible to assess more specific alterations, such as copy number variation with small panels, but also large deletion, homologous recombination deficiency, tumor mutational burden (TMB) and specific genomic signatures in the case of sequencing of tumor and germline exomes or genomes. For several years, different studies have tried to associate genomic scars with homologous recombination deficiency [23,24], yet without conclusive results. In our group of 41 patients treated with olaparib from 2015 to 2017, tumor and germline exome sequencing data were available for 14 patients, enabling analysis of complex data such as TMB, the number of neopeptides, the presence of Alexandrov’s signature 3 (Appendix A) [3,15], the number of small deletions, the number of deletions and loss of heterozygosity (LOH) > 10 Mb, and the HRD score (Appendix A, Appendix A). Based on these six complex analyses in this small patient group, we were unable to link response to olaparib with any of the above-mentioned criteria except one which was LOH > 10 Mb. Indeed, TMB, which could reflect repair defect (Figure 3A), the number of neopeptides (Figure 3B), the presence of Alexandrov’s signature 3 (Figure 3C), the number of small deletions (Figure 3D), and the HRD score (Figure 3E), all failed to predict response to olaparib in our patients. Concerning the number of large deletions or LOH > 10 Mb, it surprisingly appeared that PFS was better for patients with tumors with low genomic instability (*p* < 0.0001) (Figure 3F). Complex analyses are expensive and did not seem to detect sensitivity to olaparib, except the MyChoice test from Myriad Genetics [25]. Based on these observations, we investigated whether simple quantitative and qualitative data obtained by NGS analysis, from panel to genome, could predict response to olaparib. The first criteria assessed was the percentage of tumor cells harboring the alternative variant, and whether a loss of wild-type allele was present. The presence of a loss of the wild-type allele increases the allele frequency of the mutated allele on NGS data. To answer these questions, we generated a table allowing us to assess the percentage of tumor cells harboring the mutation, and by extension whether a loss of the wild-type allele was present (Figure 3G). The table shows the theoretical percentage of alleles harboring the variant, depending on the allelic frequency of the variant and the tumor cell content assessed by a pathologist. When the variant is heterozygous, the allelic frequency should theoretically correspond to 50% of the tumor cell content in the case of a somatic mutation. For example, in a sample containing 60% tumor cells, the allelic frequency should be around 30%. In the case of a loss of the wild-type allele, the assessment of tumor cell content based on allelic frequency should be higher than the tumor cell content obtained by a pathologist. In Figure 3G, zones in green indicate a putative loss of the wild-type allele (indicating a potential benefit from of PARP inhibitor therapy), whereas in the zones indicated in red, a loss of the wild-type allele cannot be observed (no use for a PARP inhibitor). Based on Figure 3G, we tested the potential of the second hit presence (Table 3) to predict the sensitivity of VUS-containing tumors to olaparib. It appeared that the presence of the second hit could not significantly predict PFS under olaparib when used alone (Figure 3H). In parallel, we assessed whether in silico prediction of pathogenicity could better predict sensitivity of tumors with VUS to olaparib. Using 18 different in silico prediction tools (Appendix A), analyzed thanks to the Varsome website (https://varsome.com/, accessed on 8 April 2020) using by-default parameters of each in silico tool, we tested whether predictions were correlated with response of these tumors to olaparib. Among the 18 in silico predictions (Appendix A, Appendix A), despite the absence of statistical significance, only the Dann [26] and PROVEAN [27] tools seemed to classify some VUS in patients with PFS > 120 days as deleterious and some other VUS in patients with PFS < 120 days as benign. As neither the presence of the second hit, nor in silico prediction alone was able to predict sensitivity of VUS-harboring tumors to olaparib, we tested the association between the presence of the second hit and in silico assessment. To this end, as soon as one of the predictions indicated that a variant was deleterious (the presence of the second hit and/or in silico deleterious prediction), the VUS was classified as deleterious (Table 3). In this way, we found that all VUS classified as benign were found in patients with PFS < 120 days, and the loss of wild-type allele was associated with PROVEAN (Figure 3I) or Dann (Figure 3J) predictions. As it is well known that patient Performance Status (PS) could be a marker of response to treatment, we tested whether response to olaparib could be linked to PS, regardless of the classification of the variants. As expected, PS was directly associated with PFS under olaparib, with all patients with PS3 having PFS < 120 days (Figure 3K), suggesting that this parameter could also be used before olaparib prescription. By adding this data to our model of VUS pathogenicity prediction, two PS3 patients with a VUS classified as deleterious were classified as non-responders (benign variant). We obtained a very good classification of VUS for predicting responsiveness to olaparib, with the loss of the wild-type allele being associated with the results obtained using the PROVEAN (Figure 3L) or Dann (Figure 3M) prediction tools.

### 3.4. VUS Classification Allows one to Predict Benefit from Particular PARP Inhibitors

Three PARP inhibitors are approved for the treatment of ovarian cancer: olaparib, niraparib, and rucaparib. The first is indicated in the presence of a deleterious *BRCA1* or *BRCA2* mutation, whereas the two others are indicated in all comers. From 2016 to 2020, 38 patients with an ovarian high-grade serous carcinoma were treated with either olaparib (*n* = 14) or niraparib (*n* = 24) in our institution (Table 4). In accordance with the European recommendations, all patients with a deleterious mutation in the *BRCA1* or *BRCA2* gene were treated with olaparib, except for one patient with a deleterious *RAD51C* mutation. Among patients treated with niraparib, nine had no mutation in homologous repair genes, six had a benign variant in one of the homologous repair genes (*BARD1*, *BRCA1*, *BRIP1*, *FANCD2*, *FANCF*, or *FANCG*), one had a deleterious mutation in the *BRIP1* gene, and eight patients had a VUS in one of the genes sensitizing to PARP inhibitors (*ATM*, *BRCA1*, *BRCA2*, *FANCF*, *INPP4B*, *PALB2*, *RAD50*, or *RAD51B*). Despite the fact that patients treated with olaparib had better PFS than those treated with niraparib (Figure 4A), probably because of a presence of a homologous recombination deficiency in the olaparib arm, about half (11/24) of the niraparib-treated patients had experienced some benefit from this treatment (PFS > 120 days). Based on this observation, we investigated whether a variant, regardless of its significance, could impact PFS of patients treated with niraparib. We found that the presence of a variant in genes sensitizing to PARP inhibitors did not impact PFS (Figure 4B). When we classified patients with variants by the known variant impact on protein function, it appeared that the only patient with a deleterious variant had the best PFS, whereas the majority of those with benign variants had a PFS below 120 days. Patients with VUS were equally distributed on both sides of the 120-day PFS threshold (Figure 4C). By using both our models for the prediction of pathogenicity using association of loss of the wild-type allele with Dann or PROVEAN in silico prediction tools, we tried to dichotomize niraparib-treated patients as carriers of either benign or deleterious variants (Table 5). The Dann-based model was unable to distinguish PFS curves (Figure 4D), whereas the PROVEAN-based model tended to show slightly separate curves, albeit without reaching statistical significance (Figure 4E). This could possibly be explained by the low number of patients or the efficiency of niraparib to treat non-HRD tumors. Finally, we pooled all ovarian cancer patients treated with PARP inhibitors (either olaparib or niraparib, *n* = 60) and dichotomized the population as carriers of none or a benign variant versus carriers of a deleterious variant. VUS were classified using our algorithm (Figure 4F). We confirmed that patients with a known or predicted deleterious mutation in a homologous repair gene had a marked benefit from treatment with a PARP inhibitor (*p* < 0.0001), whereas about half of the patients with no variant, or a known or predicted benign variant benefited from niraparib treatment (Figure 4G).

## 4. Discussion

The approval of PARP inhibitors in clinical practice has revolutionized the care of patients with certain types of cancer. Ovarian cancer patients were the first to benefit from PARP inhibitors [1,2,3,4,28], followed by those with breast cancer [5,6], pancreatic cancer [7], prostate cancer [8] and others [29]. The efficiency of PARP inhibitors was first related to pathogenic alterations in *BRCA1* and *BRCA2*, but homologous recombination deficiency (HRD) has since been proven to sensitize ovarian tumors to PARP inhibitors as well [28,30]. Although HRD detection, irrespective of its origin (gene methylation or alterations) would enable the prescription of PARP inhibitors to ovarian cancer patients, it is difficult to apply in routine clinical practice in some countries. Moreover, in our study, the results of complex analyses did not correlate with PFS of patients, whatever the classification of the variants (Benign, Unknown, or Pathogenic). This observation could be due to the small number of patients included in our study. It is also worth noting that the association between complex analyses data and the response of patients treated with PARP inhibitors (assessed by measuring PFS) has only been studied for the HRD score. In our study, the HRD score did not predict the PFS of patients receiving PARP inhibitor treatment. However, up to now, an association between the HRD score and the response to PARP inhibitors has only been shown for ovarian cancer patients [28,30], and has not yet been studied in other cancers, whereas in our study group, the digestive tract and breast cancers were a majority.

In routine molecular diagnosis laboratories, the use of targeted NGS panels is widespread. The size of these panels ranges from two genes (*BRCA1* and *BRCA2*) to dozens of genes directly or indirectly related to homologous recombination. A particularity of homologous recombination genes is the absence of mutation hotspots, while numerous VUS, especially missense variants, are observed. Although the number of VUS for widely studied genes (i.e., *BRCA1* and *BRCA2*) has been decreasing, some genes involved in homologous recombination have not been widely investigated to date. In our study on 41 patients, VUS constituted 33% (6/18) of *BRCA1* variants, 17% (2/12) of *BRCA2* variants and as much as 50% (5/10) of variants in other genes. Some authors have tried to set up different methods for VUS classification for the *BRCA1* [31], *BRCA2* [32], and the *PALB2* genes [33] in order to predict variant pathogenicity and the resulting susceptibility to cancer. The authors of another study developed an algorithm enabling Alexandrov’s signature 3 assessment from a target panel based on the MSK-IMPACT panel [24]. In our study, Signature 3 did not seem to predict response to olaparib. It is worth emphasizing that the use of our algorithm does not need a large targeted panel. Indeed, we tested an algorithm that is based solely on NGS data and can be used to assess tumor sensitivity to PARP inhibitors, particularly olaparib. Based on PFS for three olaparib-treated patients with tumors carrying benign variants, we considered variants as deleterious when the PFS was over 120 days. Out of 12 patients with tumors harboring VUS who were treated with olaparib, six had PFS suggestive of tumor sensitivity to olaparib. The following BRCA1 variants: p.Met18Thr (associated a PFS of 240 days), p.Ser915Phe (PFS: 368 days), and p.Gly928Val (PFS: 136 days), as well as the p.Asp219Gly PALB2 variant (PFS: 224 days), and the p.Ala195Val RAD51C variant (PFS: 210 days), seemed to be linked to tumor sensitivity to olaparib and were classified as deleterious by our algorithm. By extension, we could hypothesize that these variants could also predispose germline carriers to cancer. The last patient had a tumor with two variants (BRCA2 p.Leu1620Phe and RAD51D p.Asp110Asn) predicted to be deleterious, but we were unable to ascertain which one was responsible for the response to treatment. Among the six patients with PFS below 120 days, three had tumors carrying variants classified as benign by our algorithm, one had a variant classified as benign only with the combination of loss of wild-type allele with Dann prediction, and two had variants that were classified as deleterious. Based on the response to olaparib and in silico classification, we could hypothesize that the p.Met1710Leu and p.Glu1786Asp BRCA1 variants, BRCA2 variant p.Met990Lys, and PALB2 variant p.Glu907Lys are benign variants that do not respond to olaparib and are probably not involved in cancer susceptibility. Two patients had tumors harboring variants classified as deleterious by our algorithm (BRCA1 p.Arg841Trp and UIMC1 p.Tyr564His; BRIP1 p.Met1?) but had very short PFS (12 and 27 days, respectively). It is possible that these variants could be misclassified by our algorithm, but it should be noted that both patients, who suffered from pancreatic cancer, had a Performance Status (PS) of 3. By analyzing PFS in patients undergoing olaparib treatment according to patient PS, we found that PFS seemed to be directly associated with PS. Indeed, the better PS, the longer PFS, with a complete absence of response for PS3 patients. Moreover, our study confirmed that deleterious variants in genes other than *BRCA1* and *BRCA2* could sensitize tumors to PARP inhibitors. Indeed, we observed a response to PARP inhibitors in patients with a deleterious mutation, as described by others, in *ATM* [34], *BRIP1* [35], *INPP4B* [36], *PALB2* [35], *PTEN* [37], and *RAD51C* [38] genes. It is established that other cancer types can present HRD [39] and, consequently, can be sensitive to PARP inhibitors [40,41]. In our study, we observed that one patient with endometrial cancer harboring a deleterious *PTEN* variant and one patient with metastatic basal cell carcinoma (skin cancer) carrying a deleterious variant of *RAD51C* responded to olaparib treatment. Finally, even though some PARP inhibitors are used in all comers, we observed that the knowledge of whether a variant is benign or deleterious clearly impacts the choice of treatment and consequently the patient outcomes.

## 5. Conclusions

In conclusion, our work could be useful to guide treatment decisions regarding the use of PARP inhibitors in the management of patients with tumors harboring VUS, irrespective of the cancer site. However, this retrospective analysis included a small sample, and therefore, our findings warrant confirmation in a prospective clinical trial in order to refine the sensitivity and specificity of the algorithm.

## Figures and Tables

**Figure 1 cancers-13-03113-f001:**
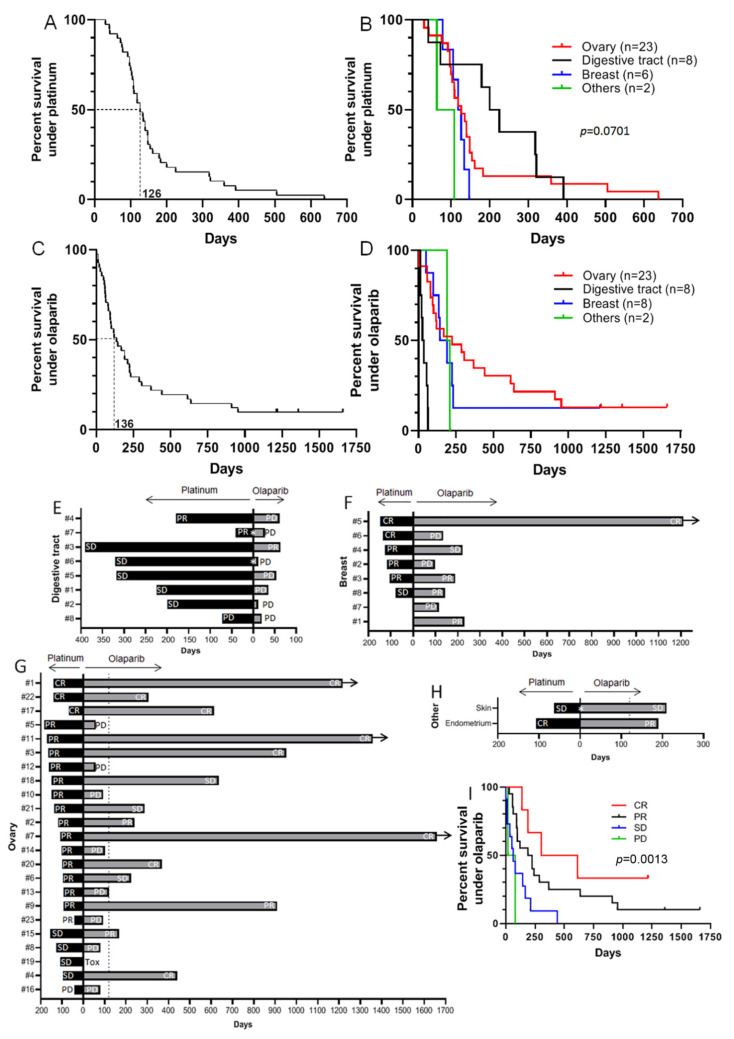
Progression-Free Survival (PFS) of patients in response to platinum salt or the PARP inhibitor olaparib. (**A**): PFS curve for 39 patients having received platinum salt-based treatment (mean PFS = 126 days). (**B**): PFS curve for patients having been administered platinum salt-based treatment, by site of origin (digestive tract: mean PFS = 212.5 days; ovary: 126 days; breast: 122 days; endometrium: 108 days; skin: 63 days). (**C**): PFS curve for 41 patients receiving olaparib treatment (mean PFS = 136 days). (**D**): PFS curves for patients receiving olaparib treatment by site of origin (digestive tract: mean PFS = 31.5 days; breast: 167 days; endometrium: 190 days; skin: 210 days; ovary: 256 days). (**E**–**H**): Duration and response to platinum treatment followed by olaparib in patients with digestive tract (**E**), breast (**F**), ovary (**G**) and other (**H**) cancers. (PD: Progressive Disease, SD: Stable Disease, PR: Partial Response, CR: Complete Response, Tox: treatment stopped due to strong toxicity). White asterisk indicates that platinum treatment and olaparib treatment were separated by another line of treatment. Black arrows indicate that olaparib treatment is still ongoing. (**I**): PFS curve for patients under olaparib treatment depending on their response to platinum-based treatment.

**Figure 2 cancers-13-03113-f002:**
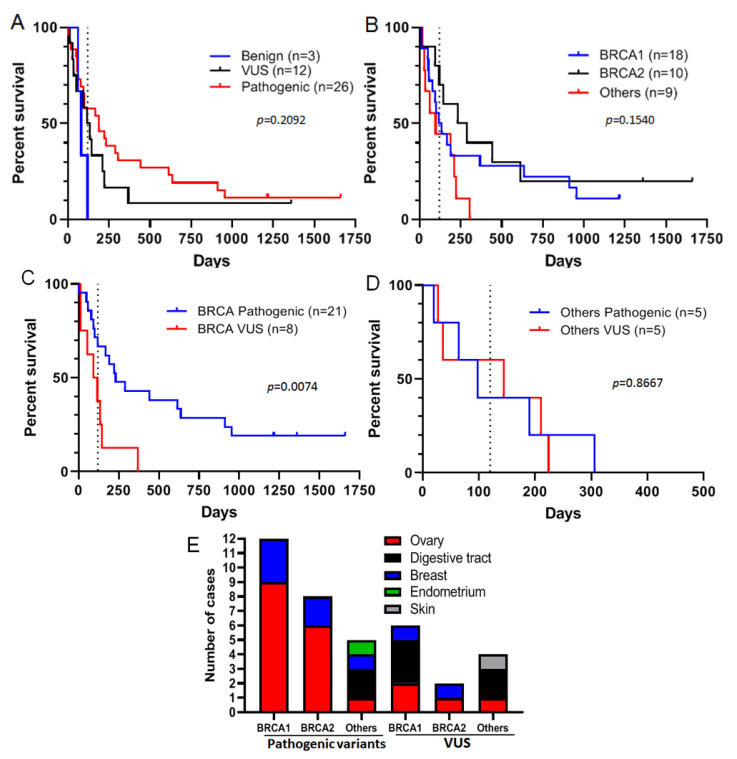
Progression-free survival (PFS) of patients under olaparib treatment according to the pathogenicity of variants detected in tumors. (**A**): PFS of patients with benign (blue), VUS (black), and pathogenic (red) variants in tumors in response to olaparib. (**B**): PFS observed for patients with tumors containing *BRCA1* (blue), *BRCA2* (black), and other gene (red) variants in response to olaparib. (**C**): PFS observed for patients with tumors containing *BRCA* pathogenic variants (blue) or BRCA VUS (red). (**D**): PFS observed for patients with tumors containing pathogenic variants (blue) or VUS (red) in other genes. (**E**): The number of pathogenic variants or VUS in *BRCA1*, *BRCA2* or other genes by site of origin. For all panels: the dashed line corresponds to 120 days.

**Figure 3 cancers-13-03113-f003:**
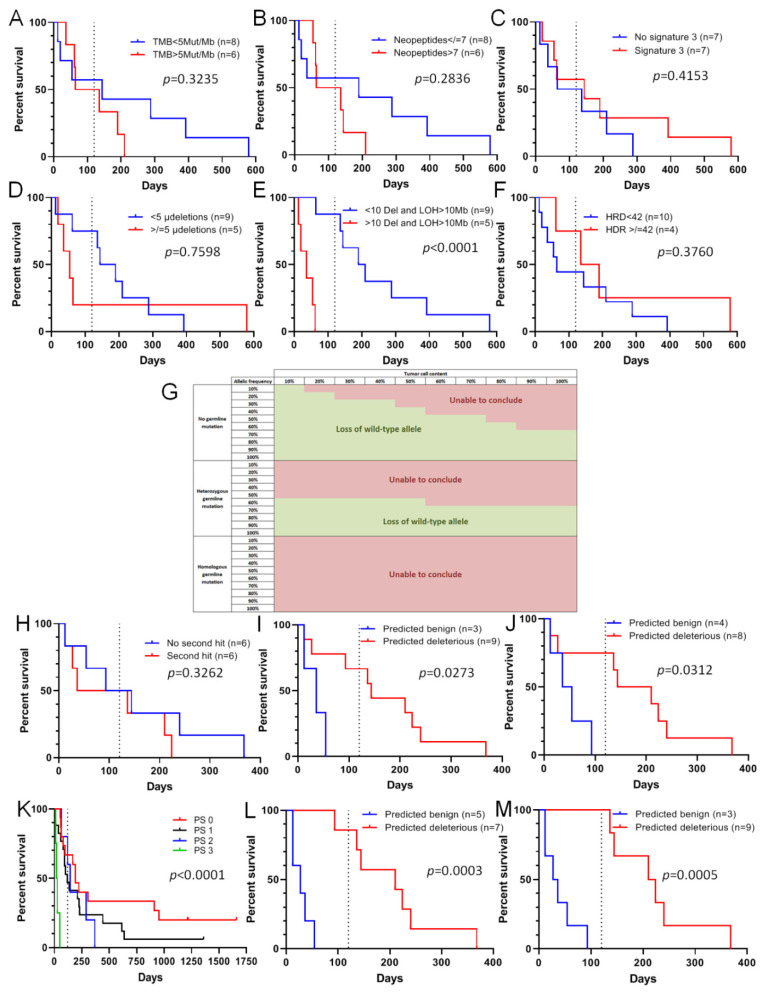
Sensitivity of VUS-containing tumors to olaparib can be assessed in silico. (**A**–**F**): Progression-free survival (PFS) for patients receiving olaparib treatment according to tumor mutational burden (TMB) (**A**), the number of neopeptides (**B**), the presence of Alexandrov’s signature 3 (**C**), the number of microdeletions (**D**), homologous recombination deficiency (HRD) score (**E**), and the number of large deletions and loss of heterozygosity (LOH) > 10 Mb (**F**,**G**): A table enabling the determination of possible LOH depending on the tumor cell content and the allele frequency of alternative variants. Green zones correspond to putative loss of the wild-type allele. (**H**–**M**). PFS under olaparib treatment according to the presence of the second hit (**H**), variant pathogenicity predicted by PROVEAN (**I**) or Dann (**J**), patient Performance Status (**K**), variant pathogenicity predicted by combination of loss of wild-type allele, Performance status and PROVEAN (**L**) or Dann (**M**) predictions. For all panels: the dashed line corresponds to 120 days.

**Figure 4 cancers-13-03113-f004:**
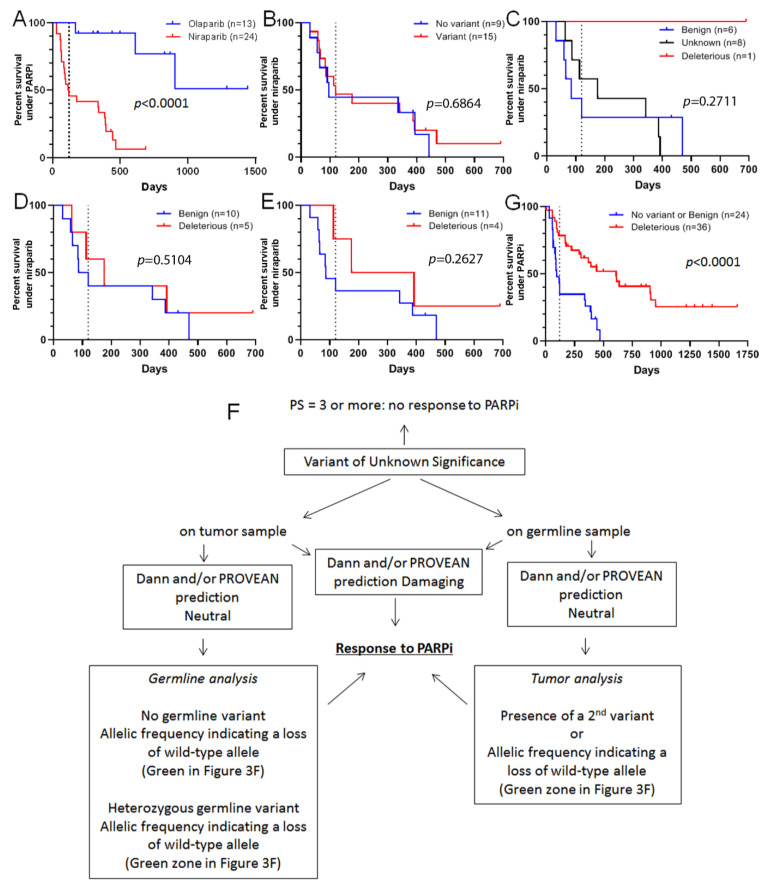
Progression-free survival (PFS) of patients with ovarian cancer treated with a PARP inhibitor. (**A**): PFS of 37 patients prospectively treated with either olaparib (blue) or niraparib (red) in compliance with the approved indication for clinical use. (**B**): PFS observed for 24 patients treated with niraparib according to the absence (blue) or presence (red) of variants, whatever their pathogenicity. (**C**): PFS observed for patients with tumors harboring benign (blue), VUS (black) and pathogenic (red) variants in response to niraparib. (**D**,**E**): PFS under niraparib treatment according to variant pathogenicity predicted by combination of loss of wild-type allele, Performance status and Dann (**D**) or PROVEAN (**E**) predictions. (**F**): A proposal of a tree to guide treatment decisions in case of VUS-containing tumors. (**G**): PFS observed for 60 patients with ovarian cancer treated with a PARP inhibitor (olaparib or niraparib) according to the absence of a variant, or the presence of a known or predicted benign variant (blue), and the presence of a known or predicted deleterious variant (red). Only statistically significant results are indicated on the graphs. For all panels: the dashed line corresponds to 120 days.

**Table 1 cancers-13-03113-t001:** Patient characteristics.

Clinical and Pathologic Characteristics	*n* (%)	Median Age (Min–Max), Years
Age at olaparib treatment	41	63 (31–84)
Organs		
Ovary	23 (56.2%)	63 (43–84)
Breast	8 (19.5%)	62 (31–83)
Digestive tract	8 (19.5%)	62 (50–76)
(pancreas, colon, rectum)	(5, 2, 1)	
Endometrium	1 (2.4%)	56
Skin	1 (2.4%)	73
Histology		
Adenocarcinoma (breast, digestive tract)	16 (39%)	
High-grade serous adenocarcinoma	22 (53.8%)	
Clear cell adenocarcinoma	1 (2.4%)	
Basal Cell carcinoma	1 (2.4%)	
Unknown	1 (2.4%)	
Response to platinum salts		Days (min–max)
Number of patients treated	39 (92.9%)	
Progression Free Survival		126 (30–637)
PFS < 90 days	7 (18%)	
PFS > 90 days	32 (82%)	

**Table 2 cancers-13-03113-t002:** List of variants observed in our population of 41 patients.

Cancer Type, Patient No.	Gene(s)	Nucleotide Variant	Protein Variant	Impact	PFS (Days)
Ovarian #1	*BRCA1*	c.798_799delTT	Ser267LysfsTer19	Pathogenic	1218 (still under olaparib)
Ovarian #2	*BRCA1*	c.53T > C	p.Met18Thr	Unknown	240
Ovarian #3	*BRCA1*	c.2477_2478delCA	p.Thr826ArgfsTer4	Pathogenic	953
Ovarian #4	*BRCA2*	c.7617 + 1G > T		Pathogenic	441
Ovarian #5	*BRCA1*	c.181T > G	p.Cys61Gly	Pathogenic	59
Ovarian #6	*PALB2*	c.656A > G	p.Asp219Gly	Unknown	224
Ovarian #7	*BRCA2*	c.3847_3848delGT	p.Val1283LysfsTer3	Pathogenic	1659 (still under olaparib)
Ovarian #8	*BRCA1*	c.3708T > G	p.Asn1236Lys	Benign	81
Ovarian #9	*BRCA1*	c.2066_2069delGTAA	p.Ser689LysfsTer11	Pathogenic	910
Ovarian #10	*BRCA1*	c.3839_3843delinsAGGC	p.Ser1280_Gln1281delinsTer	Pathogenic	94
Ovarian #11	*BRCA2*	c.8504C > G	p.Ser2835Ter	Pathogenic	1359 (still under olaparib)
Ovarian #12	*BRCA1*	c.4956G > A	p.Met1652Ile	Benign	58
	*BRCA2*	c.9976A > T	p.Lys3326Ter	Benign	
Ovarian #13	*BRCA1*	c.349C > T	p.His117Tyr	Unknown	120
	*BRCA2*	c.8494G > T	p.Glu2832Ter	Pathogenic	
Ovarian #14 *	*BRCA1*	c.4204C > T	p.Gln1402Ter	Pathogenic	101
Ovarian #15 *	*BRCA1*	c.4204C > T	p.Gln1402Ter	Pathogenic	168
Ovarian #16	*BRCA1*	c.68_69delAG	p.Glu23ValfsTer17	Pathogenic	79
Ovarian #17	*BRCA2*	c.3267_3268delGA	p.Gln1089HisfsTer9	Pathogenic	614
Ovarian #18	*BRCA1*	c.191G > A	p.Cys64Tyr	Pathogenic	636
Ovarian #19	*BRCA2*	c.5350_5351delAA	p.Asn1784HisfsTer2	Pathogenic	1 (allergic reaction)
Ovarian #20	*BRCA1*	c.2744C > T	p.Ser915Phe	Unknown	368
Ovarian #21	*BRCA2*	c.2539A > T	p. Arg847Ter	Pathogenic	288
Ovarian #22	*ATM*	c.103C > T	p.Arg35Ter	Pathogenic	306
Ovarian #23	*BRCA2*	c.1690T > C	p.Met990Lys	Unknown	93
Breast #1	*BRCA2*	c.1981_1984dup	p.Ser662Ter	Pathogenic	231
Breast #2	*BRIP1*	c.2002delG	p.Glu668LysfsTer20	Pathogenic	98
Breast #3	*BRCA1*	c.5341G > T	p.Glu1781Ter	Pathogenic	190
Breast #4	*BRCA2*	c.7654dupA	p.Ile2552AsnfsTer2	Pathogenic	223
	*BRCA2*	c.7645_7668delTGCATAAAAATTAACAGCAAAAAT	p.Cys2549_Asn2556del	Unknown	
Breast #5	*BRCA1*	c.4251_4252delG > T	p.Leu1418ArgfsTer9	Pathogenic	1212 (still under olaparib)
Breast #6	*BRCA1*	c.2783G > T	p.Gly928Val	Unknown	136
Breast #7	*BRCA1*	c.3485delA	Asp1162ValfsTer48	Pathogenic	116
Breast #8	*BRCA2*	c.4860A > T	p.Leu1620Phe	Unknown	144
	*RAD51D*	c.328G > A	p.Asp110Asn	Unknown	
Digestive tract #1 (colon)	*PALB2*	c.2719G > A	p.Glu907Lys	Unknown	36
Digestive tract #2 (pancreas)	*BRCA1*	c.2521C > T	p.Arg841Trp	Unknown	12
	*UIMC1*	c.1690T > C	p.Tyr564His	Unknown	
Digestive tract #3 (pancreas)	*CHEK2*	c.349A > G	p.Arg160Gly	Pathogenic	64
Digestive tract #4 (rectum)	*BRCA1*	c.2521C > T	p. Arg841Trp	Probably Benign	62
Digestive tract #5 (pancreas)	*BRCA1*	c.5128A > C	p.Met1710Leu	Unknown	54
Digestive tract #6 (pancreas)	*BRCA1*	c.5295A > C	p.Glu1786Asp	Unknown	12
Digestive tract #7 (pancreas)	*BRIP1*	c.3G > A	p.Met1?	Unknown	27
Digestive tract #8 (pancreas)	*ATM*	c.598C > T	p.Gln200Ter	Pathogenic	20
Endometrium #1	*PTEN*	c.867dupA	p.Val290SerfsTer8	Pathogenic	190
Skin #1	*PALB2*	c.2431C > T	p.Pro811Ser	Unknown	210
	*RAD50*	c.3041A > G	p.Gln1014Arg	Unknown	
	*RAD51C*	c.584C > T	p.Ala195Val	Unknown	

* The same patient was treated with olaparib twice. Platinum salt chemotherapy was administered before the first olaparib treatment. A second platinum salt treatment was administered before the re-challenge with olaparib. Platinum sensitivity was observed each time.

**Table 3 cancers-13-03113-t003:** Prediction of VUS pathogenicity.

Cancer Type, Patient No.	Gene(s)	Nucleotide Variant	Protein Variant	Second Hit Prediction *	Dann Prediction	PROVEAN Prediction	P S	Final Prediction	PFS (Days)
Ovarian #2	*BRCA1*	c.53T > C	p.Met18Thr	B (0.19, Red)	D	D	1	S	240
Ovarian #6	*PALB2*	c.656A > G	p.Asp219Gly	D (1.12, Green)	B	B	1	S	224
Ovarian #20	*BRCA1*	c.2744C > T	p.Ser915Phe	B (0.14, Red)	D	D	2	S	368
Ovarian #23	*BRCA2*	c.1690T > C	p.Met990Lys	B (0.26, Red)	B	D	1	S or R	93
Breast #6	*BRCA1*	c.2783G > T	p.Gly928Val	D (1.04, Green)	U	D	1	S	136
Breast #8	*BRCA2*	c.4860A > T	p.Leu1620Phe	B (0.44, Red)	B	D	2	S	144
	*RAD51D*	c.328G > A	p.Asp110Asn	B (0.16, Red)	D	D			
Digestive tract #1 (colon)	*PALB2*	c.2719G > A	p.Glu907Lys	B (0.66, Red)	B	B	1	R	36
Digestive tract #2 (pancreas)	*BRCA1*	c.2521C > T	p.Arg841Trp	D (1.48, Green)	B	D	3	R	12
	*UIMC1*	c.1690T > C	p.Tyr564His	D (0.91, Green)	D	D			
Digestive tract #5 (pancreas)	*BRCA1*	c.5128A > C	p.Met1710Leu	B (0.34, Red)	B	B	0	R	54
Digestive tract #6 (pancreas)	*BRCA1*	c.5295A > C	p.Glu1786Asp	B (0.27, Red)	U	B	1	R	12
Digestive tract #7 (pancreas)	*BRIP1*	c.3G > A	p.Met1?	D (0.96, Green)	U	U	3	R	27
Skin #1	*PALB2*	c.2431C > T	p.Pro811Ser	B (0.53, Red)	B	B	1	S	210
	*RAD50*	c.3041A > G	p.Gln1014Arg	B (0.54, Red)	B	B			
	*RAD51C*	c.584C > T	p.Ala195Val	D (0.72, Green)	D	B			

* Score values in brackets; B = Benign; D = Deleterious; U = Unknown; PS = Performance Status; S = PARPi Sensitive; R = PARPi Resistant

**Table 4 cancers-13-03113-t004:** Clinical and pathologic characteristics of ovarian cancer patients.

Characteristics	Olaparib *n* (%)	Niraparib *n* (%)
Total	14	24
Histology		
High grade serous adenocarninoma	12 (86%)	24 (100%)
Endometrioid carcinoma	2 (14%)	0
Response to platin		
Complete response	5 (36%)	8 (33%)
Partial response	7 (50%)	14 (58%)
Stable disease	2 (14%)	2 (9%)

**Table 5 cancers-13-03113-t005:** Prediction of VUS pathogenicity in a new cohort of ovary cancers.

Genes	Nucleotide Variant	Protein Variant	Second Hit Prediction	Dann Prediction	Final Prediction with PROVEAN	PS	Treatment	Final Prediction	PFS (Days)
*INPP4B*	c.1381T > C	p.Phe461Leu	D	B	N	1	Nira	S	392
*ATM*	c.2578G > C	p.Asp860His	B	D	D	1	Nira	S	176
*FANCF*	c.373G > A	p.Asp125Asn	B	B	B	1	Nira	S	113
*PALB2*	c.1273G > A	p.Val425Met	D	B	B				
*RAD51B*	c.902G > A	p.Ser301Asn	B	D	B				
*ATM*	c.4079G > A	p.Ser1360Asn	B	B	B	1	Nira	R	342
*BRCA2*	c.9109C > G	p.Gln3037Glu	U	B	B	2	Nira	R	84
*BRCA2*	c.1181A > C	p.Glu394Ala	U	B	B	1	Nira	R	87
*BRCA1*	c.1692T > A	p.Asn564Lys	B	B	B	1	Nira	R	469
*RAD50*	c.527C > T	p.Thr176Ile	U	D	B	1	Nira	U	63

B = Benign; D = Deleterious; U = Unknown; PFS: progression-free survival; PS = Performance Status; Nira = Niraparib; S = PARP inhibitors Sensitive; R = PARPi Resistant.

## Data Availability

The datasets used and/or analyzed during the current study are available from the corresponding author on reasonable request.

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
