# Peer review of "An Algorithm Combining Patient Performance Status, Second Hit Analysis, PROVEAN and Dann Prediction Tools Could Foretell Sensitization to PARP Inhibitors in Digestive, Skin, Ovarian and Breast Cancers"

_cancers, 2021, doi:10.3390/cancers13133113_

Round 1

Reviewer 1 Report

The study evaluated 78 patients with metastatic cancer who treated with PARPi. However, the heterogeneous and retrospective study design and the small sample size were insufficient to conclude.

Author Response

Review Report Form

Open Review

English language and style

( ) Extensive editing of English language and style required
(x) Moderate English changes required
( ) English language and style are fine/minor spell check required
( ) I don't feel qualified to judge about the English language and style

English will be edited a second time after manuscript improvement by a native English speaker

Yes

Can be improved

Must be improved

Not applicable

Does the introduction provide sufficient background and include all relevant references?

( )

(x)

( )

( )

Is the research design appropriate?

( )

( )

(x)

( )

Are the methods adequately described?

( )

(x)

( )

( )

Are the results clearly presented?

(x)

( )

( )

( )

Are the conclusions supported by the results?

( )

( )

(x)

( )

Comments and Suggestions for Authors

The study evaluated 78 patients with metastatic cancer who treated with PARPi. However, the heterogeneous and retrospective study design and the small sample size were insufficient to conclude.

We are disappointed if the Reviewer has found that our study is not sufficient to conclude. We agree with the fact that the sample size is small but it is important to notice that patients treated with olaparib without pathogenic BRCA variant are very rare, due to an off labeled use of the treatment. In fact, our study is a kind of proof of concept study that will allow development of clinical trials specifically studying VUS. We insisted at the end of the discussion section on the fact that our results need to be tested on a larger cohort, larger cohort that will be possible only in a specific clinical trial that needs a published proof of concept work. We hope that the improved version of the manuscript will have a more positive reception from the Reviewer.  

Reviewer 2 Report

I leave the technical details on lab procedures and how to conduct survival analyses in a multifactorial setting with multiple parameters without a a priori declared hypothesis to test to others. The statistical considerations should, however, be rewritten to more precisely warn against multiple testing in a limited sample having high probability of false positive results. 

Besides this, I have the following position after reading through the basic content of this paper:

The finding that BRCA1/2 VUSes were restricted to breast/ovarian cancer cases strongly indicates that these VUSes are pathogenic but currently misclassified as VUSes. 

The finding that carriers of these VUSes responded to Oliparib supports the above interpretation.

This is, as a life-long member of the networks having developed the classification systems for determining variants to be pathogenic or not, an important finding which should be made public asap. 

Because of the low numbers and multiple parameters used for analyses, the results in this study should be repeated by others, for which purpose publication of results is mandatory.

ENIGMA / BRCAexchange should be notified on these findings for their continous work to develop better methods for determining pathogenicity of variants.

When comprehensive genetic testing of tumours for selection of treatment now is being undertaken, one need to compile results of studies like this to interpret outcomes of the testing. 

Author Response

Open Review

English language and style

( ) Extensive editing of English language and style required
( ) Moderate English changes required
( ) English language and style are fine/minor spell check required
(x) I don't feel qualified to judge about the English language and style

Yes

Can be improved

Must be improved

Not applicable

Does the introduction provide sufficient background and include all relevant references?

(x)

( )

( )

( )

Is the research design appropriate?

(x)

( )

( )

( )

Are the methods adequately described?

(x)

( )

( )

( )

Are the results clearly presented?

(x)

( )

( )

( )

Are the conclusions supported by the results?

(x)

( )

( )

( )

We warmly thank the Reviewer for her/his review and constructive remarks. We hope that this new version of the manuscript will meet her/his requirements for an acceptance to publish our work.  

Comments and Suggestions for Authors

I leave the technical details on lab procedures and how to conduct survival analyses in a multifactorial setting with multiple parameters without a a priori declared hypothesis to test to others. The statistical considerations should, however, be rewritten to more precisely warn against multiple testing in a limited sample having high probability of false positive results. 

We agree with the Reviewer. We added a specific sentence to warn about multiple testing and the small number of cases.

Besides this, I have the following position after reading through the basic content of this paper:

The finding that BRCA1/2 VUSes were restricted to breast/ovarian cancer cases strongly indicates that these VUSes are pathogenic but currently misclassified as VUSes. 

The finding that carriers of these VUSes responded to Oliparib supports the above interpretation.

This is, as a life-long member of the networks having developed the classification systems for determining variants to be pathogenic or not, an important finding which should be made public asap. 

Because of the low numbers and multiple parameters used for analyses, the results in this study should be repeated by others, for which purpose publication of results is mandatory.

ENIGMA / BRCAexchange should be notified on these findings for their continous work to develop better methods for determining pathogenicity of variants.

When comprehensive genetic testing of tumours for selection of treatment now is being undertaken, one need to compile results of studies like this to interpret outcomes of the testing. 

We warmly thank the Reviewer who perfectly understands our problematic and the impact of our work.

Reviewer 3 Report

Dear Authors,

here attached are my comments. I hope they will be useful to improve your manuscript.

Summary

This paper aims to contribute to solve a pressing problem. A rational approach indicates that not only the already categorized BRCA1/2 loss of function mutants but also many still notcategorized homologous recombination (HR) mutants could benefit from a PARP inhibitor therapy (PARPi). However, up to date only few identified BRCA1/2 mutants are engaged in PARPis therapy because of a lack of tools to pick the remnant dysfunctional HR patients. Two main approaches are being undertaken by the scientific community to solve this problem: (1) genomic scars due to dysfunctional HR are being characterized and (2) meanwhile, HR genes variants of unknown significance (VUS) are intended to being classified as soon as possible. Here, a retrospective analysis is carried on 41 patients carrying VUS, pathogenic or benign mutations who underwent PARPis therapy (39 of them, after platinum therapy) with variable outcomes expressed as progression free survival days (PFS). In a cohort of 14 VUS patients for whom tumor and germline exomes data were available, type (1) analysis (including Tumor mutation burden (TBM), number of neopeptides, Alexandrov’s signature, number of small deletions, number of deletions and LOH >10 Mb and HRD score), would not have predicted the outcome. Then, type (2) analysis was assessed. Three algorithms (second hit prediction, Dann and PROVEAN) were used to attempt to predict the pathogenicity of each VUS. It is claimed that these combined algorithms plus patient performance status (PS) would have predicted the outcome. If I understood correctly, then the proposed criteria would be: “If one of the three algorithms predicts that the VUS is deleterious and the patient performance status (PS) is <3, PARPis will work”. This criteria was tested in a new cohort of patients, predicting correctly 6/8 outcomes (75 %, see Table 4), which is quite an improvement. The authors want to promote further testing of this prediction in larger samples.

Broad comments:

In the introduction, please introduce the main principles behind second hit, Dann and PROVEAN (in silico prediction tools or algorithms chosen from Varsome website).

Please complete the Materials and Methods section in a way that allows understanding and reproduction.

Please re-consider the prediction tree (Figure 4F).

Specific comments

Regarding the methods:

Please explicit the by-default or chosen assumptions (for example, PROVEAN cutoff is 2.5 by default, but it can be changed to get either better sensitivity or better specificity; have you taken this into consideration?)

The difference among PROVEAN vs “our PROVEAN” (figure 3 I vs L) or Dann vs “our Dann” (Figure 3 J vs M) is not clear. In Tables 3 and 4, does “final prediction with Dann” refer to “our Dann” and “final prediction with PROVEAN” refer to “our PROVEAN”? In line 247 it says: “We obtained very good classification of VUS for response to olaparib with association of LOH and PROVEAN (Figure 3L) or Dann (Figure 3M) prediction tools”. How were LOH and PROVEAN or Dann associated or combined?

Importantly, if the prediction of PARPi sensitivity algorithm implies using genomic LOH information (aside that from the VUS under consideration), then the title of the manuscript should be changed because it cannot be distinguished whether it is the specific VUS or a broader genomic context what makes the patient sensitive to PARPi. If that is the case, the title could be something like: “In low performance status scored cancer patients, certain variants of unknown significance in homologous recombination genes, in a particular LOH context, were predicted to sensitize to PARPis”.

LOH was determined according to tumor cell content and allele frequency of alternative variants in the germline. Can you further explain the method? How is Table 3G read? . What does it mean for example 10 % tumor cell content and 100 % no germline mutation, corresponds to 1000 % LOH? Or 10 % tumor cell content and 10 % homologous germline mutation corresponds to -800 % LOH? How were figures 3 E, L and M obtained?

Figure 4 F (the decision tree) is important and is not clear. Please re-build this figure. If Performance Status (PS) is so easy to assess and PS ≥ 3 is an exclusion criteria, why is it not the first item to be taken into consideration in the decision tree? What would be the reason to carry complex studies to realize in the end that the patient has to be excluded due to high PS, which can be assessed by looking at the patient or asking him/her a few questions? In my opinion, the decision tree should start there. Then, according to Table 3 and Table 4, other items should be the “final prediction with Dann”, the “final prediction with PROVEAN” or the second hit prediction. With PS <3, if either of these three assessments leads to a “deleterious variant” prediction, then the outcome with PARPis should be favourable. However, your decision tree in Figure 4 F does not refer to “our Dann” or “final prediction with Dann”, it just says “Dann”. Are you referring to the one in Varsome website or the one you modified by ‘association” with LOH? The same happens regarding PROVEAN. And “green in figure 3F” is not a clearly stated criteria.

Other comments

In the introduction, it could be briefly explained that PARP enzymatic inhibition and PARP trapping are mechanisms behind PARPis effects, and a reference on the relative capacity of trapping of olaparib and niraparib (see for example Kim et al 2021 at https://doi.org/10.1038/s12276-021-00557-3 or https://doi.org/10.3390/ijms22084203) could be included.

Rapid figure interpretation (figures 1 -4) would benefit from including the therapy agent in the PFS graph ordinates, eg: Percent survival under olaparib.

In Tables 3 and 4 the words Deleterious or Neutral could be substituted by a letter in the intermediate columns, so that there would be place for an extra column stating the final-final prediction: PARPi-sensitive or PARPi-resistant, so that it could be directly compared to the obtained results (PFS).

In the discussion, could you further contrast your prediction method with SigMA, by

Gulhan et al 2019 (https://doi.org/10.1038/s41588-019-0390-2) ?

Author Response

Open Review

English language and style

( ) Extensive editing of English language and style required
( ) Moderate English changes required
(x) English language and style are fine/minor spell check required
( ) I don't feel qualified to judge about the English language and style

English will be edited a second time after manuscript improvement by a native English speaker

Yes

Can be improved

Must be improved

Not applicable

Does the introduction provide sufficient background and include all relevant references?

( )

( )

(x)

( )

Is the research design appropriate?

( )

(x)

( )

( )

Are the methods adequately described?

( )

( )

(x)

( )

Are the results clearly presented?

( )

( )

(x)

( )

Are the conclusions supported by the results?

( )

(x)

( )

( )

Comments and Suggestions for Authors

Dear Authors,

here attached are my comments. I hope they will be useful to improve your manuscript.

We thank the Reviwer for her/his constructive comments.

Summary

This paper aims to contribute to solve a pressing problem. A rational approach indicates that not only the already categorized BRCA1/2 loss of function mutants but also many still notcategorized homologous recombination (HR) mutants could benefit from a PARP inhibitor therapy (PARPi). However, up to date only few identified BRCA1/2 mutants are engaged in PARPis therapy because of a lack of tools to pick the remnant dysfunctional HR patients. Two main approaches are being undertaken by the scientific community to solve this problem: (1) genomic scars due to dysfunctional HR are being characterized and (2) meanwhile, HR genes variants of unknown significance (VUS) are intended to being classified as soon as possible. Here, a retrospective analysis is carried on 41 patients carrying VUS, pathogenic or benign mutations who underwent PARPis therapy (39 of them, after platinum therapy) with variable outcomes expressed as progression free survival days (PFS). In a cohort of 14 VUS patients for whom tumor and germline exomes data were available, type (1) analysis (including Tumor mutation burden (TBM), number of neopeptides, Alexandrov’s signature, number of small deletions, number of deletions and LOH >10 Mb and HRD score), would not have predicted the outcome. Then, type (2) analysis was assessed. Three algorithms (second hit prediction, Dann and PROVEAN) were used to attempt to predict the pathogenicity of each VUS. It is claimed that these combined algorithms plus patient performance status (PS) would have predicted the outcome. If I understood correctly, then the proposed criteria would be: “If one of the three algorithms predicts that the VUS is deleterious and the patient performance status (PS) is <3, PARPis will work”. This criteria was tested in a new cohort of patients, predicting correctly 6/8 outcomes (75 %, see Table 4), which is quite an improvement. The authors want to promote further testing of this prediction in larger samples.

Broad comments:

In the introduction, please introduce the main principles behind second hit, Dann and PROVEAN (in silico prediction tools or algorithms chosen from Varsome website).

In the new version, we try to improve principles of second hit, and in silico predictions.

Please complete the Materials and Methods section in a way that allows understanding and reproduction.

In the new version, we complete Materials and Methods section.

Please re-consider the prediction tree (Figure 4F).

We agree with the Reviewer and change Figure 4F in the new version of the manuscript.

Specific comments

Regarding the methods:

Please explicit the by-default or chosen assumptions (for example, PROVEAN cutoff is 2.5 by default, but it can be changed to get either better sensitivity or better specificity; have you taken this into consideration?)

We used by-default parameters to obtain the better equilibrium between sensitivity and specificity. Indeed, PROVEAN by-default cutoff is -2.5 to obtain sensitivity of 80.4 and specificity of 78.6. If we decrease this threshold up to -4.1, we effectively increase specificity to 90.3 but sensitivity decreases to 57.6. To the contrary, if we increase the threshold to -1.3, the sensitivity goes up to 90.6 but the specificity goes down to 61.6. We added in the new version a sentence indicating that e used by-default parameters.

The difference among PROVEAN vs “our PROVEAN” (figure 3 I vs L) or Dann vs “our Dann” (Figure 3 J vs M) is not clear.

‘’PROVEAN’’ in Figure 3I corresponds to only in silico PROVEAN prediction of pathogenicity whereas in Figure 3L, we indicated ‘’our algorithm using PROVEAN’’, that means that we use presence of a second hit, in silico prediction with PROVEAN and Performance Status.

In Tables 3 and 4, does “final prediction with Dann” refer to “our Dann” and “final prediction with PROVEAN” refer to “our PROVEAN”?

In Table 3 and 4, ‘’final prediction with’’ refers to the use of both presence of a second hit plus in silico prediction with either Dann or PROVEAN’’. In this case, Performance Status was not used.

In line 247 it says: “We obtained very good classification of VUS for response to olaparib with association of LOH and PROVEAN (Figure 3L) or Dann (Figure 3M) prediction tools”. How were LOH and PROVEAN or Dann associated or combined?

This point was explained in the text from line 235 to line 239: ‘’As neither presence of a second hit, nor in silico prediction was able to predict alone sensitivity of VUS to olaparib, we tested the association between presence of a second hit and in silico assessment. To this end, as soon as one of the predictions indicated deleterious (presence of a second hit and/or in silico deleterious prediction) the VUS was classified as deleterious (Table 3)’’. In order to avoid misunderstanding, we have replaced LOH by presence of a second hit in the new version of the manuscript.

Importantly, if the prediction of PARPi sensitivity algorithm implies using genomic LOH information (aside that from the VUS under consideration), then the title of the manuscript should be changed because it cannot be distinguished whether it is the specific VUS or a broader genomic context what makes the patient sensitive to PARPi. If that is the case, the title could be something like: “In low performance status scored cancer patients, certain variants of unknown significance in homologous recombination genes, in a particular LOH context, were predicted to sensitize to PARPis”.

We understand the comment of the Reviewer. As what we named LOH was in fact presence of a second hit on the homologous recombination gene carrying a VUS mutation, and as we modified the text in the new version, we do not think that it is necessary to change the title of the manuscript. Nevertheless, if the Reviewer and the Editor want this modification, we will do it.

LOH was determined according to tumor cell content and allele frequency of alternative variants in the germline. Can you further explain the method? How is Table 3G read? . What does it mean for example 10 % tumor cell content and 100 % no germline mutation, corresponds to 1000 % LOH? Or 10 % tumor cell content and 10 % homologous germline mutation corresponds to -800 % LOH? How were figures 3 E, L and M obtained?

We want to apologize for the misunderstanding related to the use of the term ‘’LOH’’ in the section related to Figure (Table) 3G. Indeed, the use of LOH is confusing between genome wide LOH detection (as in Figure 3E and detected by the analysis of tumor exome in combination with germline exome) and loss of the wild-type allele in a homologous repair gene (as in Figure 3G). To avoid this kind of misunderstanding, we replaced the term ‘’LOH’’ by ‘’loss of the wild-type allele’’ for homologous recombination genes and kept LOH only for genome wide LOH. Figure 3G enables the detection of a loss of the wild-type allele by using allele frequency of the VUS from tumor NGS analysis (exome or smaller panels). We agree that the Figure is confusing with values > 100%. In order to clarify, we have suppressed the percentage and just kept red and green colors. We indicated in red sections “unable to conclude” and in the green sections “loss of wild-type allele”. We also induced an explanation in the Figure legend of panel 3G to understand the germline mutation subsets.

Figure 4 F (the decision tree) is important and is not clear. Please re-build this figure. If Performance Status (PS) is so easy to assess and PS ≥ 3 is an exclusion criteria, why is it not the first item to be taken into consideration in the decision tree? What would be the reason to carry complex studies to realize in the end that the patient has to be excluded due to high PS, which can be assessed by looking at the patient or asking him/her a few questions? In my opinion, the decision tree should start there. Then, according to Table 3 and Table 4, other items should be the “final prediction with Dann”, the “final prediction with PROVEAN” or the second hit prediction. With PS <3, if either of these three assessments leads to a “deleterious variant” prediction, then the outcome with PARPis should be favourable. However, your decision tree in Figure 4 F does not refer to “our Dann” or “final prediction with Dann”, it just says “Dann”. Are you referring to the one in Varsome website or the one you modified by ‘association” with LOH? The same happens regarding PROVEAN. And “green in figure 3F” is not a clearly stated criteria.

We totally agree with the Reviewer and warmly thank her/him for her/his remarks. We tried to modify the figure in order to make it more readable and understandable.

Other comments

In the introduction, it could be briefly explained that PARP enzymatic inhibition and PARP trapping are mechanisms behind PARPis effects, and a reference on the relative capacity of trapping of olaparib and niraparib (see for example Kim et al 2021 at https://doi.org/10.1038/s12276-021-00557-3 or https://doi.org/10.3390/ijms22084203) could be included.

We added a brief explanation and the Kim et al reference.

Rapid figure interpretation (figures 1 -4) would benefit from including the therapy agent in the PFS graph ordinates, eg: Percent survival under olaparib.

We modified PFS graphs as asked for Figures 1 and 4.

In Tables 3 and 4 the words Deleterious or Neutral could be substituted by a letter in the intermediate columns, so that there would be place for an extra column stating the final-final prediction: PARPi-sensitive or PARPi-resistant, so that it could be directly compared to the obtained results (PFS).

We modified Tables 3 and 4 in accordance with Reviewer’s comments.

In the discussion, could you further contrast your prediction method with SigMA, by

Gulhan et al 2019 (https://doi.org/10.1038/s41588-019-0390-2) ?

We completed the discussion section with this point.

Reviewer 4 Report

The review is attached

The paper investigates the possibility to offer PARPi in case of UV in HRR genes.

The manuscript needs extensive English revision

Major revisions:

Introduction:

Lane 50: the pathogenic or likely pathogenetic variant meaning is mostly related to functional assays by immunoprecipitation, western blotting, gamma irradiation and comparative structural modeling, that need a lot of time to be performed. Add  this methodology to cosegregation in the family.

Materials and methods:

Describe how were patients selected in this study. There was an ethical commitment approval for this experimental study? How clear cell carcinoma or colo-rectal cancer or basal skin cell carcinoma were choosen for PARPi treatment?

Was the basal cell carcinoma of the skin treated by platinum-derived drugs? It’s a very strange situation, since no distant metastasis are usually seen in this kind of tumor. Was the patient affected from any other tumor? Put all the patients characteristics in a table, adding stage, grading and previous treatment before PARPi

Results

Do you mean that 2 patients received in first line PARPi without chemotherapy before? Better explain who were and why did not receive chemotherapy before.

Put in the Figure 1 the PFS data beside the curves and in the legend

How were the MMR genes in colo-rectal cancer? Olaparib is not indicated in case of mutation in MMR genes.

Looking at the pancreatic cancer, I’ve noted that none BRCA2 mutation was found but only BRCA1, ATM or CHEK2 genes appeared mutated.  Furthermore, also a probably benign mutation in a rectal cancer patient was inserted. It’s quite expected that Olparib did not provide any effect in terms of response duration.

The Olaparib response is not only related to platinum-derived drugs. Also anthracyclines and cyclofosfamide can induce a double strand break DNA, more than platinum-derived drugs. You need to add all administered treatments before PARPi in order to show if any alkylant or topoisomerse inhibitors were provided, particularly in case of breast cancer.

In regards to Olaparib response, data are not enough to provide subclassification among different genes, VUS, benign and pathogenic variants. Since BRCA1/2 genes are similar and more represented than others, I suggest to merge figures 2 C/D in only one, in order to improve the differences between pathogenetic and vUS variants.

In the “Response of VUS to olaparib could be predicted before treatment” section, I appreciate the attempt to correlate the PS to Olaparib response and VUS classification. Patients with PS3 are mostly treated with palliative care: so it’s not surprising that Olaparib do not work in these patients It should be of interest if other parameters could be analyzed in a multivariate analysis. In case of ovarian cancer, the Ca.125 value can be used as predictor of response/pathogenetic variants. Other variables can be grading, stage and other treatent before PARPi. Provide a tabel at this aim.

I can’t find the reference to Table 4 in the text. What does it mean?

Author Response

Open Review

English language and style

(x) Extensive editing of English language and style required
( ) Moderate English changes required
( ) English language and style are fine/minor spell check required
( ) I don't feel qualified to judge about the English language and style

English will be edited a second time after manuscript improvement by a native English speaker

Yes

Can be improved

Must be improved

Not applicable

Does the introduction provide sufficient background and include all relevant references?

( )

(x)

( )

( )

Is the research design appropriate?

( )

( )

(x)

( )

Are the methods adequately described?

( )

( )

(x)

( )

Are the results clearly presented?

( )

(x)

( )

( )

Are the conclusions supported by the results?

(x)

( )

( )

( )

Comments and Suggestions for Authors

The review is attached

The paper investigates the possibility to offer PARPi in case of UV in HRR genes.

The manuscript needs extensive English revision

Major revisions:

Introduction:

Lane 50: the pathogenic or likely pathogenetic variant meaning is mostly related to functional assays by immunoprecipitation, western blotting, gamma irradiation and comparative structural modeling, that need a lot of time to be performed. Add  this methodology to cosegregation in the family.

We thank the Reviewer for her/his remarks. We agree with this point and added this methodology.   

Materials and methods:

Describe how were patients selected in this study. There was an ethical commitment approval for this experimental study? How clear cell carcinoma or colo-rectal cancer or basal skin cell carcinoma were choosen for PARPi treatment?

As mentioned in the Declarations section, patients gave their consent to participate and ethics approval was obtained because the study was related to the EXOMA clinical trial. We added a specific section in the materials and methods section and cited EXOMA related article.

Was the basal cell carcinoma of the skin treated by platinum-derived drugs? It’s a very strange situation, since no distant metastasis are usually seen in this kind of tumor. Was the patient affected from any other tumor? Put all the patients characteristics in a table, adding stage, grading and previous treatment before PARPi

We agree with the Reviewer that metastatic basal cell carcinoma is a very rare situation. We are currently writing a case report concerning this case that happens in 1 case on 10000. We performed exome analyses of the primary BCC and the metastatic lesion and we obtained the same mutational profile for both lesions. As indicated from lane 143 to 148 and in Figure 1H of the previous version of the manuscript, basal cell carcinoma was treated by platinum salt with a stable disease obtained. Concerning patients’ characteristics, it is indicated lane 98 that all patients were metastatic and previous treatments before olaparib were indicated for patients from lanes 125 to 148, and each patient appeared in Figure 1E-H.

Results

Do you mean that 2 patients received in first line PARPi without chemotherapy before? Better explain who were and why did not receive chemotherapy before.

We imagine that the Rewiewer is talking about the 2 patients with metastatic breast cancer indicated in Figure 1F. Indeed, both patients did not receive platinum salt but, as indicated lanes 133-134-135 of the previous version, they received olaparib after anthracycline based treatment. That is why no response to platinum was mentioned in Figure 1F.

Put in the Figure 1 the PFS data beside the curves and in the legend

We added PFS data mentioned in the text beside the curves for panel A and C and in the legend for all panels.

How were the MMR genes in colo-rectal cancer? Olaparib is not indicated in case of mutation in MMR genes.

As all colorectal cancers had an exome analysis, MMR status was systematically studied. For all patients treated with olaparib, MMR status was normal. In case of MMR deficiency, patients would have benefit from a national clinical trial using immunotherapy.

Looking at the pancreatic cancer, I’ve noted that none BRCA2 mutation was found but only BRCA1, ATM or CHEK2 genes appeared mutated.  Furthermore, also a probably benign mutation in a rectal cancer patient was inserted. It’s quite expected that Olparib did not provide any effect in terms of response duration.

In our small cohort of pancreatic cancer, we did not observe any BRCA2 mutation. The Reviewer is right, one patient with a benign variant was treated with PARPi, as 2 other patients (ovaries #8 and 12). The choice of treatment was done by physicians. As mentioned in the manuscript, these data were used to have a reference of no response to PARPi. 

The Olaparib response is not only related to platinum-derived drugs. Also anthracyclines and cyclofosfamide can induce a double strand break DNA, more than platinum-derived drugs. You need to add all administered treatments before PARPi in order to show if any alkylant or topoisomerse inhibitors were provided, particularly in case of breast cancer.

We agree with the Reviewer concerning the fact that not only platinum derived treatments induce double strand break of DNA. To answer to Reviewer’s comment, and to avoid overloading the manuscript, we just added the following sentence for breast cancer: ‘’after several lines with anthracyclins, taxanes, 5-fluorouracil, and eribulin’’.

In regards to Olaparib response, data are not enough to provide subclassification among different genes, VUS, benign and pathogenic variants. Since BRCA1/2 genes are similar and more represented than others, I suggest to merge figures 2 C/D in only one, in order to improve the differences between pathogenetic and vUS variants.

We have merged Figure 2C and 2D for classification of BRCA pathogenic mutations.

In the “Response of VUS to olaparib could be predicted before treatment” section, I appreciate the attempt to correlate the PS to Olaparib response and VUS classification. Patients with PS3 are mostly treated with palliative care: so it’s not surprising that Olaparib do not work in these patients It should be of interest if other parameters could be analyzed in a multivariate analysis. In case of ovarian cancer, the Ca.125 value can be used as predictor of response/pathogenetic variants. Other variables can be grading, stage and other treatent before PARPi. Provide a tabel at this aim.

The goal of our work is to classify a VUS before treatment start. Consequently, CA125 cannot be used, in our case, as a predictor of response since its value can be predictive of response once the treatment started. In our work, we studied only metastatic patients (stage IV) and showed that PS3 patients did not respond to PARPi treatment. We also showed that response to platinum was significantly correlated with PFS under olaparib. We think that we have already given all parameters that could predict response to PARPi. To our point of view, adding another table would overload the manuscript.  

I can’t find the reference to Table 4 in the text. What does it mean?

We apologize for this mistake. Indeed, we have forgotten this citation. We have now fixed this mistake and renamed Table 5 in Table 4, and Table 4 in Table 5 and cited the new Table 5 in the manuscript.

Round 2

Reviewer 3 Report

Now I understood there was confusion among genomic LOH vs allelic second hit. Thank you.

Importantly, I still think that the title does not match what your work shows. Your work has not proved, for any of the VUS, that a new VUS sensitizes to PARPi by itself. It is proposing instead that, in patients with a certain performance status (which may depend on other factors besides the analyzed genes carrying a VUS), certain VUS in HR genes predict a sensitization to PARPi. An alternative title could be: “An algorithm combining patient performance status, second hit analysis, PROVEAN & Dann prediction tools could foretell sensitization to PARP inhibitors in digestive, skin, ovarian and breast cancers”. The word “could” is important, because only after trying the algorithm in larger cohorts will it be proven if the algorithm really works.

___________-

You answered: ‘’PROVEAN’’ in Figure 3I corresponds to only in silico PROVEAN prediction of pathogenicity whereas in Figure 3L, we indicated ‘’our algorithm using PROVEAN’’,that means that we use presence of a second hit, in silico prediction with PROVEAN and Performance Status.In Table 3 and 4, ‘’final prediction with’’ refers to the use of both presence of a second hit plus in silico prediction with either Dann or PROVEAN’’. In this case, Performance Status was not used.

This is very confusing. Please, leave the terms PROVEAN and Dann to the pure in silico predictions. In tables, include your algorithm ingredients (PROVEAN, Dann, second hit and PS) and then a single combined “final prediction”, the one of your algorithm.

The final prediction is only one, dependent on four algorithm ingredients (performance status, second hit analysis, PROVEAN & Dann prediction). In tables 3 and 4, can you please erase the word “final” in “final prediction with PROVEAN” and “final prediction with Dann”? Just inform the in silico result. The reader can combine it with the second hit and with PS, leading to your algorithm prediction.

______________

In table 5, is the prediction match for PARP inhibitor Niraparib 4 hits in 8 cases? If PFS=120, then S(392), S(176), R(84) and R (87)  are correct predictions. On the other hand, R(469), R (342), S(63) and S(113) are incorrect predictions. 50 % prediction is random…This means that the prediction did not work in the new cohort. Am I wrong?

Author Response

Open Review

English language and style

( ) Extensive editing of English language and style required
( ) Moderate English changes required
(x) English language and style are fine/minor spell check required
( ) I don't feel qualified to judge about the English language and style

Yes

Can be improved

Must be improved

Not applicable

Does the introduction provide sufficient background and include all relevant references?

(x)

( )

( )

( )

Is the research design appropriate?

(x)

( )

( )

( )

Are the methods adequately described?

(x)

( )

( )

( )

Are the results clearly presented?

( )

(x)

( )

( )

Are the conclusions supported by the results?

( )

( )

(x)

( )

Comments and Suggestions for Authors

Now I understood there was confusion among genomic LOH vs allelic second hit. Thank you.

We thank the Reviewer for having helped us to correct the confusion. 

Importantly, I still think that the title does not match what your work shows. Your work has not proved, for any of the VUS, that a new VUS sensitizes to PARPi by itself. It is proposing instead that, in patients with a certain performance status (which may depend on other factors besides the analyzed genes carrying a VUS), certain VUS in HR genes predict a sensitization to PARPi. An alternative title could be: “An algorithm combining patient performance status, second hit analysis, PROVEAN & Dann prediction tools could foretell sensitization to PARP inhibitors in digestive, skin, ovarian and breast cancers”. The word “could” is important, because only after trying the algorithm in larger cohorts will it be proven if the algorithm really works.

We modified the title as proposed by the Reviewer.

___________-

You answered: ‘’PROVEAN’’ in Figure 3I corresponds to only in silico PROVEAN prediction of pathogenicity whereas in Figure 3L, we indicated ‘’our algorithm using PROVEAN’’, that means that we use presence of a second hit, in silico prediction with PROVEAN and Performance Status. In Table 3 and 4, ‘’final prediction with’’ refers to the use of both presence of a second hit plus in silico prediction with either Dann or PROVEAN’’. In this case, Performance Status was not used.

This is very confusing. Please, leave the terms PROVEAN and Dann to the pure in silico predictions. In tables, include your algorithm ingredients (PROVEAN, Dann, second hit and PS) and then a single combined “final prediction”, the one of your algorithm.

The final prediction is only one, dependent on four algorithm ingredients (performance status, second hit analysis, PROVEAN & Dann prediction). In tables 3 and 4, can you please erase the word “final” in “final prediction with PROVEAN” and “final prediction with Dann”? Just inform the in silico result. The reader can combine it with the second hit and with PS, leading to your algorithm prediction.

As proposed by the Reviewer, we modified the text, figure legends and Tables 3 and 4 to avoid confusion between PROVEAN and Dann in silico predictions and our algorithm.

______________

In table 5, is the prediction match for PARP inhibitor Niraparib 4 hits in 8 cases? If PFS=120, then S(392), S(176), R(84) and R (87)  are correct predictions. On the other hand, R(469), R (342), S(63) and S(113) are incorrect predictions. 50 % prediction is random…This means that the prediction did not work in the new cohort. Am I wrong?

The cohort treated with Niraparib is rather difficult to interpret. Indeed, as mentioned in the Results section, Niraparib has shown efficiency in HRD positive tumors but also in HRD negative tumors, suggesting that even if a variant is benign (really classified or predicted), the patient can experience a response to this PARP inhibitor.  Consequently, a variant predicted benign with a PFS superior to 120 days does not mean that the prediction is wrong.

To the contrary, when a variant is predicted as deleterious, PFS should be superior to 120 days. In Table 4, we have a variant classified S with a PFS of 113 days, that is very close to the limit of 120 days but we agree that this case is not a positive result. It is nevertheless important to know that in some cases, PARP inhibitors do not work event in presence of a pathogenic variant. That is why we think that it is important to perform a clinical trial in order to confirm our work on a greater population.

The variant with a PFS of 63 days was really difficult to classify as we were not able to predict a second hit and Dann and PROVEAN had contradictory predictions. In the Table, we decided to indicate Unknown for this variant.  

Reviewer 4 Report

The manuscript has been improved as recommended and is suitable for publication in Cancers

Author Response

Open Review

English language and style

( ) Extensive editing of English language and style required
( ) Moderate English changes required
(x) English language and style are fine/minor spell check required
( ) I don't feel qualified to judge about the English language and style

Yes

Can be improved

Must be improved

Not applicable

Does the introduction provide sufficient background and include all relevant references?

(x)

( )

( )

( )

Is the research design appropriate?

(x)

( )

( )

( )

Are the methods adequately described?

(x)

( )

( )

( )

Are the results clearly presented?

(x)

( )

( )

( )

Are the conclusions supported by the results?

(x)

( )

( )

( )

Comments and Suggestions for Authors

The manuscript has been improved as recommended and is suitable for publication in Cancers

We warmly thank the Reviewer for her/his response and acceptation and for her/his remarks that improved the manuscript.

Round 3

Reviewer 3 Report

I am glad the work has been improved. I hope the proposed algorithm will be soon tested in a larger cohort.